

# Temperature (208-318 K) and pressure (18-696 Torr) dependent rate coefficients for the reaction between OH and HNO₃.

Katrin Dulitz[1,a], Damien Amedro[1], Terry J. Dillon[1,b], Andrea Pozzer[1], and John N Crowley[1]

[1] Division of Atmospheric Chemistry, Max-Planck-Institut für Chemie, 55128 Mainz, Germany
[a] now at: Nanophysics research group, University of Freiburg, 79104 Freiburg, Germany
[b] now at: Department of Chemistry, University of York, York, U.K

*Correspondence to*: John Crowley (john.crowley@mpic.de)

**Abstract.** Rate coefficients ($k_5$) for the title reaction were obtained using pulsed laser photolytic generation of OH coupled to its detection by laser-induced fluorescence (PLP-LIF). More than eighty determinations of $k_5$ were carried out in nitrogen or air bath gas at various temperatures and pressures. The accuracy of the rate coefficients obtained was enhanced by in-situ measurement of the concentrations of both $HNO_3$ reactant and $NO_2$ impurity. The rate coefficients show both temperature and pressure dependence with a rapid increase in $k_5$ at low temperatures. The pressure dependence was weak at room temperature but increased significantly at low temperatures. The entire dataset was combined with selected literature values of $k_5$ and parameterised using a combination of pressure dependent and independent terms to give an expression that covers the relevant pressure and temperature range for the atmosphere. A global model, using the new parameterisation for $k_5$ rather than those presently accepted, indicated small but significant latitude and altitude dependent changes in the $HNO_3$ / $NOx$ ratio of between -6% and +6%. Effective $HNO_3$ absorption cross sections (184.95 and 213.86 nm, units of $cm^2$ molecule$^{-1}$) were obtained as part of this work: $\sigma_{213.86} = 4.52^{+0.23}_{-0.12} \times 10^{-19}$ and $\sigma_{184.95} = 1.61^{+0.08}_{-0.04} \times 10^{-17}$.

## 1 Introduction

Nitric acid, ubiquitous to the Earth's atmosphere, is formed mainly in the association reaction between nitrogen dioxide and hydroxyl radicals (R1) and is an important reservoir species for $NOx$ (where $NOx$ is defined as the sum of NO and $NO_2$), especially at higher altitudes, e.g. in the lower stratosphere where it represents ~ 80-100% of reactive nitrogen oxides ($NOy$ = $NOx$ + $HNO_3$ + PAN + $2N_2O_5$ etc.). Recent laboratory studies (Butkovskaya et al., 2007; Butkovskaya et al., 2009) suggest that $HNO_3$ may also be formed (at < 1% yield) in the reaction of $HO_2$ with NO, which can double the $HNO_3$ production rate in e.g. the tropical tropopause (R2) (Cariolle et al., 2008). The reaction of nitrate radicals ($NO_3$) with some organic trace gases (RH) including aldehydes and dimethylsulphide can be a significant direct source of $HNO_3$ at night (R3), as can the heterogeneous hydrolysis of both $N_2O_5$ and organic nitrates, $RONO_2$, (R4) in conjunction with gas-liquid partitioning. As $NO_3$, $N_2O_5$ and $RONO_2$ are all formed in the atmosphere via oxidation of NO and $NO_2$, these processes indirectly convert $NOx$ to $HNO_3$.



| NO$_2$ + OH + M | → | HNO$_3$ + M | (R1) |
| HO$_2$ + NO + M | → | HNO$_3$ + M | (R2) |
| NO$_3$ + RH | → | HNO$_3$ + R | (R3) |
| N$_2$O$_5$ or RONO$_2$ + H$_2$O($l$) | → | HNO$_3$ + other products | (R4) |

Reaction with the hydroxyl radicals (OH, R5) and photolysis (R6) are the major gas-phase HNO$_3$ loss processes:

| HNO$_3$ + OH | → | NO$_3$ + H$_2$O | (R5) |
| HNO$_3$ + $h\nu$ | → | OH + NO$_2$ | (R6) |

which result in atmospheric lifetimes of several weeks in the troposphere and lower stratosphere. Due to its high solubility, wet and dry deposition reduce the lifetime of HNO$_3$ in the planetary boundary layer to a few days. The nitrate radical product

of R5 (NO$_3$) is rapidly converted to either NO$_2$ and NO by photolysis or to NO$_2$ by reaction with NO, so that (R5) represents an important route for re-activation of NO$x$ from the long-lived HNO$_3$ reservoir. Traditionally, atmospheric models have tended to over-predict nitric acid concentrations and under-predict NO$x$ to HNO$_3$ ratios, with the largest discrepancy found at high altitudes. The model-measurement discrepancy may be related to unknown (or more rapid) re-noxification processes, either gas-phase or heterogeneous (see e.g. Thakur et al. (1999) and Schultz et al. (2000) and references therein for a

summary) or due to difficulties in modelling NO$x$ input from lightning and HNO$_3$ scavenging (Staudt et al., 2003). Given the major role of HNO$_3$ re-noxification in determining e.g. O$_3$ production rates in the atmosphere, it is of major importance to reduce the uncertainty in the rate coefficient for the title reaction.

The important atmospheric role of the reaction between OH and HNO$_3$ is reflected in the numerous experimental determinations of the rate coefficient ($k_5$) and the products as listed and reviewed by evaluation panels (Atkinson et al., 2004;

Sander et al., 2006). Most of the experimental studies were carried out in the mid-70s to mid-80s (Margitan et al., 1975; Smith and Zellner, 1975; Wine et al., 1981; Jourdain et al., 1982; Kurylo et al., 1982; Margitan and Watson, 1982; Marinelli and Johnston, 1982; Ravishankara et al., 1982; Devolder et al., 1984; Smith et al., 1984; Connell and Howard, 1985; Jolly et al., 1985; Stachnik et al., 1986) with only the most recent and comprehensive studies of the reaction (Brown et al., 1999; Brown et al., 2001) extending the temperature range to those found at the tropopause. Experimental (Carl et al., 2001;

McCabe et al., 2003; O'Donnell et al., 2008b) and theoretical (Xia and Lin, 2001; Gonzalez and Anglada, 2010) work examining the details of the reaction mechanism highlight continuing interest in this complex reaction, which proceeds via formation of a pre-reaction complex, HO-HNO$_3$ (Aloisio and Francisco, 1999; Brown et al., 1999; Brown et al., 2001; Xia and Lin, 2001; O'Donnell et al., 2008a). HO-HNO$_3$ can (a) dissociate into reactants, (b) rearrange to form products via a transition state which lies somewhat higher in energy than the reactants or (c) experience collisional deactivation by bath gas

molecules. Dissociation of the thermalised complex into the NO$_3$ and H$_2$O reaction products is via tunnelling through the exit barrier, which explains the unusual kinetics observed, with $k_5$ increasing with pressure and decreasing temperature (Margitan and Watson, 1982; Stachnik et al., 1986; Brown et al., 1999). No evidence for products apart from NO$_3$ and H$_2$O has been obtained (Atkinson et al., 2004).





As mentioned above, prior to the present dataset, only Brown et al. (1999) had conducted experiments under conditions of temperature and pressure relevant for the tropopause / lower stratosphere (< 240 K). The experiments of Brown at al. revealed a rapid increase in $k_5$ at low temperatures, leading to a reduced $NO_2$ to $HNO_3$ ratio in models of this part of the atmosphere. The important findings of Brown et al. require confirmation from independent experiments and the present

study is intended to provide highly accurate rate coefficients which cover sufficient parameter space to do this.

## 2 Experimental

### 2.1 Pulsed laser photolysis, laser induced fluorescence set-up

Rate coefficients ($k_5$) for the title reaction were determined using pulsed laser photolysis (PLP) coupled to laser-induced fluorescence detection (LIF) as illustrated in Fig. 1. A detailed description of the PLP-LIF apparatus has been published

(Wollenhaupt et al., 2000) and only essential details and modifications are reproduced here. The central component is a quartz reactor, the temperature of which was controlled by circulating a cryogenic fluid through an outer jacket. The inner wall of the reactor was coated with a thin film of Teflon (DuPont, FEP TE9568) to reduce adsorption of $HNO_3$. The temperature at the intersection of the laser beams (defining the reaction volume) was monitored by a thermocouple before and after each experimental series. All kinetic measurements were carried out under "slow flow conditions" with

approximate linear flow velocities of 10 cm s$^{-1}$ preventing the build-up of reaction products inside the reaction volume. The reactor pressure was monitored using calibrated 100 Torr and 1000 Torr capacitance manometers.

### 2.2 Generation and detection of OH

A KrF exciplex laser (Lambda Physik Lextra 50), operated at λ = 248 nm, was used to produce OH radicals from $HNO_3$ which served both as reactant and precursor (R6). Hydroxyl radicals were excited by a Nd-YAG pumped dye laser

(rhodamine 6G, Lambda Physik Scanmate) at λ = 281.915 nm ($A^2\Sigma^+$ (ν' = 1) ← $X^2\Pi$(ν'' = 0), $Q_1$(1) transition). OH fluorescence from the electronic $A^2\Sigma^+$ (ν' = 0) → $X^2\Pi$(ν'' = 0) transition at λ ≈ 309  nm was collected by a $MgF_2$ lens located perpendicular to the laser beams and detected by a photomultiplier tube (PMT) connected to a gated boxcar integrator. An interference filter (309 ± 5 nm) and a BG 26 glass cut-off filter were used to decrease scattered light at the PMT. The exciplex laser fluence was measured by a calibrated Joule meter located at the end of the beam axis in order to

estimate OH concentrations, [OH]. The exciplex laser fluence was adjusted so that the [$HNO_3$]/[OH] ratio was always > 10$^4$ ensuring first-order conditions and suppressing interfering secondary OH chemistry (see section 3.3).

### 2.3 HNO$_3$ flows and concentration measurement

Frequently, in a study of this sort where pseudo-first order kinetics are anticipated, the main source of error in deriving the rate constant is the measurement of the concentration of the excess reagent ($HNO_3$ in this case) and any impurities that may




also react with OH. For this reason, much effort was dedicated to the accurate, in-situ measurement of $HNO_3$ concentrations using spectroscopic methods and also optical detection (and reduction) of potential impurities.

Gas-phase $HNO_3$ was eluted into the reactor in a flow of $N_2$ which was passed over a liquid $HNO_3$ reservoir prior to mixing and dilution with bath gas. Three different liquid $HNO_3$ sources were used: a ternary mixture (50 wt.% $HNO_3$ / 22 wt.%

$H_2SO_4$), anhydrous $HNO_3$, and 90% $HNO_3$. The ternary mixture was used only in experiments at $T > 263$ K due to $HNO_3$ condensation inside the photolysis reactor at lower temperatures. The anhydrous $HNO_3$ sample was found to contain traces of $NO_3$ as observed previously (Crowley et al., 1993). These could be significantly reduced by the addition of $H_2O$ to generate the 90 wt.% sample.

### 2.3.1 HNO₃ concentration measurement using absorption spectroscopy

Nitric acid concentrations were determined optically using dual beam absorption cells located downstream of the photolysis reactor. The first absorption cell ($l = 43.8$ nm) was equipped with a low pressure, Hg "Penray" lamp to measure the attenuation of light by $HNO_3$ at 184.95 nm (Wollenhaupt et al., 2000) and was used mainly at $T \leq 227$ K where low $HNO_3$ concentrations were used. For experiments above 227 K, $HNO_3$ absorption at 213.86 nm in the second optical absorption cell ($l = 34.8$ cm) equipped with a low-pressure zinc lamp (213.86 nm) and ($214 \pm 10$) nm bandpass filter (LOT-Oriel) was used

to determine [$HNO_3$]. For the 184.95 nm measurements the transmitted light intensity was corrected for detection of "wrong" wavelengths (e.g. 253.65 nm) by adding large concentrations of gases ($N_2O$ or $CCl_4$) which attenuate the 184.95 line completely. This method of measurement of nitric acid effectively integrates its concentration throughout the reactor and, if wall losses of $HNO_3$ result in significant radial gradients in its concentration, may not necessarily give a representative measure of [$HNO_3$] at the centre of the reactor, especially when the reactor was cold. For this reason, we also measured the

$HNO_3$ concentration the centre of the reactor (in the same volume in which OH decays were recorded) using a two-photon excitation scheme which is summarised briefly below.

### 2.3.2 HNO₃ Concentration Measurement via two-photon excitation at 193 nm

The detection of $HNO_3$ via two-photon excitation at 193 nm (Two Photon, Excited Fragment Spectroscopy, TPEFS) was

pioneered by Stuhl and co-workers and used to measure $HNO_3$ in ambient air (Papenbrock et al., 1984; Kenner et al., 1985; Kenner et al., 1986; Papenbrock and Stuhl, 1991) We shall present a detailed description of the detection scheme and the photo-physics involved and the application of TPEFS to kinetic studies in a separate publication and simply outline the central features here. $HNO_3$ is converted in a sequential, two-photon process to an electronically excited hydroxyl radical, OH(A), the fluorescence of which can be detected at 308 nm. Excited (triplet state) HONO is believed to be formed upon

absorption of the first 193 nm photon and dissociated to OH(A) and NO(X,A) by the second photon. Gently focussing the 193 nm beam from an ArF Excimer laser to a diameter of about 1 mm was sufficient to obtain a detection limit of $\approx 10^8$ molecule cm$^{-3}$ at 50 Torr of $N_2$ bath gas.



The TPEFS detection of $HNO_3$ was calibrated by flowing pure samples of $HNO_3$ through the reactor, and monitoring its concentration by absorption spectroscopy at 184.95 nm.

## 2.4 On-line Determination of Nitrogen Dioxide Concentrations

A major potential systematic error in measuring the rate coefficient for $OH + HNO_3$ is the presence of $NO_2$, an unavoidable impurity in $HNO_3$ samples, which reacts rapidly with OH, especially at high pressures and low temperatures. $NO_2$ is formed in the heterogeneous decomposition of gaseous $HNO_3$ on surfaces and, to a lesser extent, due to the liquid-phase disproportionation of anhydrous $HNO_3$ and thermal decomposition of $N_2O_5$ (R7 and R8) (Crowley et al., 1993).

| | | | |
|---|---|---|---|
| 2 $HNO_3$ | $\rightarrow$ | $N_2O_5 + H_2O$ | (R7) |
| $N_2O_5$ | $\rightarrow$ | $NO_2 + NO_3$ | (R8) |

$NO_2$ impurity levels were measured in-situ using a multi-pass absorption cell ($l$ = 880 cm) positioned downstream of the LIF reactor and the other optical cells. Light from a halogen lamp passing through the cell was focused onto the entrance slit of a 0.5 m monochromator (B&M Spektronik BM50, 600 lines $mm^{-1}$ grating blazed at 500 nm). A diode-array detector (Oriel Instaspec 2) was used to record $NO_2$ absorption between 398 and 480 nm. The instrumental resolution ($\delta\lambda$= 0.32 nm) was determined from the full width at half maximum (FWHM) of the 435.8 nm Hg emission line.

## 2.5 Determination of NO, NO$_3$ and HONO impurities

The same multi-pass absorption cell was used to monitor $NO_3$ (592-671 nm) and HONO (253-335 nm) impurities. NO concentrations were estimated using a 30.4 cm absorption cell with a $D_2$ lamp providing analysis light between 177 and 260 nm. For this measurement, the monochromator (600 lines $mm^{-1}$ grating blazed at 200 nm) was purged with nitrogen. The instrumental resolution was set at 0.32 nm (FWHM of a Hg line) in every experiment.

## 2.6 HNO$_3$ Absorption Cross Sections at 213.86 nm and 184.95 nm

The 213.86 nm optical absorption setup was used to determine an effective $HNO_3$ absorption cross section at this wavelength. In these experiments, a flow of undiluted, anhydrous nitric acid was passed through the 34.8 cm absorption cell with concentrations calculated from the $HNO_3$ pressure, which was measured with a 2 Torr capacitance manometer. The accuracy of the pressure measurement was confirmed by cross-checking with 0.1 Torr and 10 Torr manometers. The temperature of the cell was monitored with a thermocouple. Light intensity in the absence of $HNO_3$ was obtained by evacuating the absorption cell and purging with $N_2$. An absorption cross section at 184.95 nm was derived by connecting the 184.95 and 213.86 nm optical absorption cells in series and measuring relative optical densities at both wavelengths for the same sample.



## 2.7 Chemicals

Anhydrous nitric acid was prepared by mixing $KNO_3$ (Merck) with $H_2SO_4$ (95-98 wt.%, Roth) and condensing the $HNO_3$ vapour into a liquid nitrogen trap. The ternary mixture (50 wt.% $HNO_3$ / 22 wt.% $H_2SO_4$) was made from $H_2SO_4$ (95-98 wt.%, Roth) and $HNO_3$ (65 wt.%, Roth). All nitric acid sources were stored at $T = 253$ K between experiments. Apart from

experiments in $N_2$ bath gas (Westfalen, 5.0) a few measurements were conducted in air (approx. 79% $N_2$, 21% $O_2$) by mixing $N_2$ carrier gas with $O_2$ (Air Liquide, 4.5).

## 3 Results and Discussion

### 3.1 $HNO_3$ Absorption Cross Sections at 213.86 nm and 184.95 nm

The $HNO_3$ absorption cross section at 213.86 nm, $\sigma_{213.86\,nm}$, was determined by measuring optical density while varying the

$HNO_3$ pressure between 0.125 Torr and 2.019 Torr. $HNO_3$ concentrations were varied either by adjusting the valves at the entrance or at the exit of the absorption cell or by increasing the temperature of the $HNO_3$ reservoir, both methods giving the same result. The cross section determinations are summarised in Fig. 2 where measured optical densities at 213.86 nm ($OD_{213.86}$) are related to the $HNO_3$ concentration and the cross section, $\sigma_{213.86}$, via the Beer-Lambert expression:

$$OD_{213.86} = \ln\left(\frac{I_0}{I}\right) = \sigma_{213.86\,nm} \cdot l \cdot [HNO_3] \qquad \text{(i)}$$

where $l$ is the cell length (34.8 cm) and $I_0$ and $I$ are the intensities of incident and transmitted light at 213.86 nm, respectively. A linear regression yields a value of $\sigma_{213.86\,nm} = 4.52 \times 10^{-19}$ $cm^2$ molecule$^{-1}$, whereas a proportional fit yields $\sigma_{213.86\,nm} = 4.59 \times 10^{-19}$ $cm^2$ molecule$^{-1}$ (both at 300 K). The small difference (less than 1.5%) is caused by an offset of 0.01 in optical density, which may be due to $HNO_3$ adsorbed on optical windows of the absorption cell. The total uncertainty in $\sigma_{213.86}$ was assessed by considering the difference in cross section obtained from the linear and proportional fits, the

estimated error in cell length ($\Delta l = 0.1$ cm) and the statistical error (0.1%) from the linear fit. In addition, the contribution from impurities $N_2O_5$ (0.2%), $NO_2$ (0.2%), $NO_3$ (0.01%) and $H_2O$ (2%) was found to contribute a lower limit of 1.2% and an upper limit of 2.0% to the error estimate for $\sigma_{213.86\,nm}$. The final result is thus $\sigma_{213.86nm} = 4.52^{+0.23}_{-0.12} \times 10^{-19}$ $cm^2$ molecule$^{-1}$.

An effective $HNO_3$ cross section at 184.95 nm, $\sigma_{184.95\,nm}$, was obtained by measuring the optical densities in both absorption cells simultaneously. The different cell lengths and small differences in the pressure and temperature between the cells were

accounted for in calculating $[HNO_3]$. As can be seen from the inset in Fig. 2, a linear relationship was observed, which resulted in an $HNO_3$ absorption cross section (at 300 K) of $\sigma_{184.95\,nm} = 1.61^{+0.08}_{-0.04} \times 10^{-17}$ $cm^2$ molecule$^{-1}$. As the relative cross section is very accurately defined, the errors in the cross section at 184.95 nm stem almost entirely from those in the absolute value at 213.86 nm. $HNO_3$ cross sections at these wavelengths have been previously reported. Brown et al. (1999) quote a 213.86 nm absorption cross section of $(4.52 \pm 0.19) \times 10^{-19}$ $cm^2$ molecule$^{-1}$, obtained by interpolating the $HNO_3$ absorption



spectrum of Burkholder et al. (1993) This value agrees well with that obtained in the present experiment. A very similar value of $\sigma_{183.95}$ nm = $1.63 \times 10^{-17}$ cm$^2$ molecule$^{-1}$ was first determined by (Biaume, 1973-1974) and confirmed by Wine et al. (1981) and Connell and Howard (1985) The excellent agreement with previous results indicates that our absorption cross sections are well-suited for the in-situ optical determination of HNO$_3$ concentrations.

**3.2 HNO$_3$ Detection using two-photon excitation at 193 nm**

The measurement of the HNO$_3$ concentration downstream of the vessel in which the reaction takes place may lead to a systematic bias in the rate constant if HNO$_3$ is partitioned significantly to surfaces. Uptake of HNO$_3$ to the reactor wall will result in radial gradients in the reactor, especially at high pressures where mixing by radial diffusion is slow. Similar to the study of Brown et al. (1999) we observed significant loss of HNO$_3$ from the gas-phase at low temperatures, presumably the

result of condensation on the cold reactor walls. We would then expect that the HNO$_3$ concentration close to the walls is lower than that in the centre of the reactor, implying that any method of HNO$_3$ concentration measurement that integrates over the entire reactor diameter will generate rate constants that are too high at low temperatures. This applies equally to the present study, where HNO$_3$ is measured downstream after turbulent mixing of gas leaving the reactor and also to the experiments of Brown et al. (1999) who monitored the HNO$_3$ concentration across the diameter of the reaction vessel as well

as downstream in a separate optical absorption cell.

We used the TPEFS technique outlined in Section 2.3 to measure the HNO$_3$ concentration in the centre of the reactor where OH was measured in the kinetic experiments and compared it with that measured in online absorption cells (184.9 and 213.9 nm) located downstream of the reactor. For consistency, we used similar flows, pressures and HNO$_3$ concentrations to those used for determining $k_5$. In order to simplify the analysis, most of the experiments were conducted at constant gas densities

and [HNO$_3$]. In some experiments where the nitric acid concentrations could not be kept constant (mainly at low temperature and high [HNO$_3$]), we corrected the TPEFS signal for 193 nm light absorption and HNO$_3$ quenching of the OH fluorescence. Details of the corrections are provided in the supplementary information (Fig. S1-S3).

The TPEFS signal and HNO$_3$ concentration (184.95 nm absorption) were recorded at a high temperature and the reactor was subsequently cooled to the next (lower) temperature with the pressure adjusted to keep the same molecular density. In

complementary experiments, the HNO$_3$ concentrations and densities were varied from $7.0 \times 10^{13}$ to $3.0 \times 10^{16}$ molecule cm$^{-3}$ and $1.6 \times 10^{18}$ to $4.4 \times 10^{18}$ molecule cm$^{-3}$, respectively. Figure 3 shows the normalised TPEFS signal as a function of temperature. The signal was normalised to the average [HNO$_3$] / TPEFS ratio obtained at the temperatures above 0 °C where effects of HNO$_3$ condensation were negligible. A ratio lower than unity thus indicates that use of the 184.95 nm optical absorption measurements to measure [HNO$_3$] can result in an underestimation of the HNO$_3$ concentration in the middle of

the reactor compared to the usual online measurement using the absorption cell downstream of the reactor. At the lowest temperatures (T<233 K), we observed some deviation (10 ± 10% at 208 K), indicating a possible underestimation of the HNO$_3$ concentration, but within the uncertainty of the measurement. The increase in the uncertainty in TPEFS sensitivity at





low temperatures is related to experimental problems keeping $HNO_3$ constant at these temperatures: We conclude that the effects of $HNO_3$ concentration gradients are < 10% under our experimental conditions and add an additional 10% uncertainty for the kinetic measurements at temperatures below 240 K.

### 3.3 Impurities

As the reaction between $NO_2$ and OH (R1) is pressure and temperature dependent and the rate coefficients are considerably larger than for (R5), even small amounts of $NO_2$ can contribute significantly to the OH decay measured in a study of $k_5$. Fortunately, $NO_2$ possesses a distinct, structured absorption spectrum with sufficiently large differential absorption cross sections to make accurate concentration measurements at low impurity levels possible. $NO_2$ optical densities over the 880 cm optical path length were measured online using a multi-pass cell located downstream of the photolysis reactor. $NO_2$ concentrations were calculated by least-squares fitting to a reference spectrum (Bogumil et al., 2003) (see Fig. 4). The limit of detection (LOD) was $[NO_2] \approx 2 \times 10^{11}$ molecule cm$^{-3}$. The $NO_2$ content depended upon the $HNO_3$ source used and was observed to decrease in the order anhydrous $HNO_3$ > ternary mixture > 90% $HNO_3$. No difference in the $[HNO_3]/[NO_2]$ ratio was observed when the absorption cell was relocated upstream of the LIF reactor. In general, the $NO_2$ impurity levels were low, with an $HNO_3 / NO_2$ ratio > $10^4$ so that $NO_2$ measurement was only possible at high $HNO_3$ concentrations, e.g., at room temperature. $NO_2$ is also generated as co-product in the photolysis of $HNO_3$ to make OH (R6). In this case the $[HNO_3]/[OH]$ ratio was usually also $10^4$, making this a comparable (and thus also negligible) source of $NO_2$. Using literature rate coefficients (Atkinson et al., 2004; Sander et al., 2006) for the reaction between $NO_2$ and OH (1) a contribution of R1 to the OH loss rate could be calculated from each individual decay. This was always less than 1% of the total measured OH decay rate constant and less than 0.2% at low pressures ($p \leq 50$ Torr). The use of relatively low OH concentrations ($10^{11}$ molecule cm$^{-3}$) also makes the contribution of OH self-reactions ($k_{OH+OH} \approx 4 \times 10^{-12}$ cm$^3$ molecule$^{-1}$ s$^{-1}$ at 298 K and 200 Torr) (Atkinson et al., 2004) negligibly small.

Operating under conditions of low conversion of $HNO_3$ is important for reduction of the impact of rapid secondary reactions of OH with e.g. $NO_3$ (formed in R5) or with $HO_2$ (formed in the reaction of OH with $NO_3$). Although the OH decay rate can be significantly enhanced by secondary reactions when $[HNO_3]/[OH] \leq 1000$, for $[HNO_3]/[OH] > 10^4$ this is avoided. This highlights an important advantage of real-time (e.g. pulsed/flash photolysis) experiments on (R5) compared to flow tubes, which are restricted by issues of spatial resolution in the amount of $HNO_3$ that can be employed. The main advantage is however the possibility to explore a greater spread of temperatures and pressures and to work under essentially wall free conditions, which is important when dealing with OH and $HNO_3$ which both have high affinities for surfaces.

The multi-pass absorption cell was also used to check for $NO_3$ and HONO. Although $NO_3$ absorption was indeed observed with the anhydrous nitric acid source, the mixing ratio relative to $HNO_3$ was very low, i.e., $[NO_3]/[HNO_3] < 10^{-5}$. Through the addition of $\approx 10\%$ $H_2O$ (i.e. using the 90% $HNO_3$ solution) the $NO_3$ concentration was further reduced by more than an order of magnitude. At these levels, $NO_3$ does not influence the OH decay rates by more than 0.1%. $NO_3$ absorption bands were not observed when using the ternary mixture.





We were unable to detect HONO or NO absorption features in any of the HNO$_3$ sources. Detection limits, were $\approx 5\times10^{13}$ molecule cm$^{-3}$ for NO and $\approx 2\times10^{13}$ molecule cm$^{-3}$ for HONO. At these levels, the impact on the OH decay rate or HNO$_3$ concentration measurement at 213.86 nm is negligible, e.g., an NO impurity level of $\approx 3\%$ would bias the quantitative HNO$_3$ absorption measurements at 213.86 nm by less than 2%.

## 3.4 Rate Coefficients ($k_5$) for HNO$_3$ + OH

All kinetic experiments were carried out under pseudo-first-order conditions, i.e. [HNO$_3$] >> [OH] so that the OH decay can be described as follows:

$$\ln\left(\frac{[OH]_t}{[OH]_o}\right) = -\left(k_5[HNO_3] + k_1[NO_2] + k_d\right)\cdot t = -k'\cdot t \qquad \text{(ii)}$$

where [OH]$_0$ and [OH]$_t$ are the initial and time dependent [OH] concentrations (proportional to the OH-LIF signal),
respectively. The pseudo first-order rate coefficient $k'$ comprises the rate coefficients for the target reaction HNO$_3$ with OH ($k_5$), the reaction between NO$_2$ and OH ($k_1$) and diffusion out of the reaction zone ($k_d$). A typical series of OH-LIF profiles is illustrated in Fig. 5. The OH decay was found to be strictly mono-exponential, so that a pseudo-first order rate coefficient $k'$ could be calculated from the slope of each curve. Values of $k'$ were corrected for the term $k_1[NO_2]$ to yield $k^{cor}$ and $k_5$ was obtained from the slopes of plots of $k^{cor}$ versus [HNO$_3$] (Fig. 6). The intercept is due to transport and diffusion processes, i.e.
$k_d$ and was typically ~100 s$^{-1}$ at $p$ = 20 Torr and close to zero at $p$ > 200 Torr.

Altogether, more than eighty determination of $k_5$ were made at various temperatures and pressures, in N$_2$ and air bath gases and using three different HNO$_3$ sources. Neither the source of HNO$_3$ nor the identity of the bath gas had a measurable influence on the rate coefficients obtained. Due to efficient quenching of OH fluorescence by O$_2$, most measurements were conducted in nitrogen bath gas instead of synthetic air to optimise the signal quality. Rate coefficients obtained in air (at 275
and 297 K) agreed within 4% with those obtained in N$_2$, which is consistent with the results of earlier studies (Stachnik et al., 1986; Brown et al., 1999).

The temperature and pressure inside the photolysis cell were varied over as large a range as possible (208 – 318 K and 18-696 Torr), with the lower temperature limit determined by HNO$_3$ condensation. At temperatures lower than 208 K, large fluctuations in the measured optical density of HNO$_3$ were evidence of strong partitioning to the walls of the reactor, with
periodic modulation of gas-phase [HNO$_3$] arising via weak temperature cycling ($\pm$ 0.5 °C) of the cryostat. The HNO$_3$ concentration changed by up to 50% during these experiments and gave rise to inaccurate kinetic data and scattered plots of $k'$ versus [HNO$_3$]. For this reason we report no data below 208 K.

Our measurements reveal a strong temperature and pressure dependence of $k_5$ under some conditions. The rate coefficients obtained in N$_2$ are summarised in Fig. 7 and listed in Table S1 (supplementary information). Note that data of similar quality
obtained at 239 and 246 K are not plotted in Fig. 7 to preserve clarity of presentation. The error bars (total uncertainty) include 2 σ statistical uncertainty as derived from fits to the data as in Fig. 3 and 4) and an estimate of systematic error. At




room temperature and for pressures up to ~300 Torr, the overall uncertainty in $k_5$ depends mainly on the [$HNO_3$] measurements and we assign a value of 7% to cover both uncertainty in the cross sections at 214.86 and 184.95 nm and additional uncertainty due to slight drifts in measurement of optical density at 213.86 nm. At low temperatures, the total uncertainty increased as more $HNO_3$ was stored on the walls of the reactor (leading to periodic modulation of [$HNO_3$]) and

due to the use of a restricted range of [$HNO_3$] (to keep the condensation problem manageable). Thus at 208 K and 217 K a further 10% was added to the overall uncertainty, increasing it to $\approx$ 17%.

Over the range of temperature and pressure covered in these experiments, $k_5$ varied over a factor of ~6. Although a positive trend in $k_5$ with bath gas pressure is evident at 297 and 276 K, the dependence is very weak and is only observable due to the high precision of the dataset. At temperatures of 257 K and below, the pressure dependence is more pronounced.

The temperature dependence of $k_5$ at two constant bath gas densities ($8.9 \times 10^{17}$ and $2.1 \times 10^{19}$ molecule $cm^{-3}$, corresponding to pressures of $\approx$ 27 and $\approx$ 650 Torr, respectively) is illustrated in Fig. 8. The negative dependence of $k_5$ on temperature is very clear in both cases. The data at the lower pressure are reproduced by a simple Arrhenius expression: $k_5$ (M = $8.88 \times 10^{17}$ molecule $cm^{-3}$) = $3.29 \times 10^{-15}$ exp(1079/$T$) $cm^3$ molecule$^{-1}$ s$^{-1}$. At the higher pressure curvature is more evident, though the expression $k_5$ ([M] = $2.1 \times 10^{19}$ molecule $cm^{-3}$) = $2.29 \times 10^{-15}$ exp (1221/$T$) $cm^3$ molecule$^{-1}$ s$^{-1}$ captures most of the data points.

Only at pressures much lower than those achievable with the present set-up does the temperature dependence show a significant change in the apparent (negative) activation energy with e.g. a value of $k_5$ = $2 \times 10^{-14}$ exp (430/$T$) $cm^3$ molecule$^{-1}$ s$^{-1}$ obtained in a few Torr of He (Connell and Howard, 1985).

The solid fit-lines in Fig. 7 were obtained by a least-squares fit to the present data-sets using the expression derived by (Lamb et al., 1984) and which has been used on several occasions for parameterising $k_5$:

$$k_5 = k_0(T) + k_p(M,T) = k_0 + \frac{k_c[M]}{1 + k_c[M]/k_\Delta}$$    (iii)

Here, the overall rate coefficient, $k_5$, is comprised of a pressure-independent rate coefficient $k_0(T) = A_0 \exp(E_0/T)$ and a Lindemann-Hinshelwood type term, $k_p$ (M,T), for the pressure dependence. $k_c = A_c \exp(E_c/T)$ is the termolecular rate coefficient for formation of the thermalised complex, $k_\Delta = A_\Delta \exp(E_\Delta/T)$ is $k_\infty - k_0$, the difference between the high- and low-pressure limiting rate coefficients ($k_\infty$ and $k_0$, respectively) and [M] is the bath gas concentration. As our data do not define

the rate coefficient close to the low-pressure limit, we guided the six-parameter fit (equation iii) by adding data points at M = $2.5 \times 10^{16}$ molecule $cm^{-3}$ calculated using $k_5$ = $2 \times 10^{-14}$ exp(430/$T$) $cm^3$ molecule$^{-1}$ s$^{-1}$ from the low pressure study of Connell and Howard (1985). This is represented by the vertical, solid line in Fig. 7, which covers temperatures between 220 and 330 K. The low pressure dataset of Connell and Howard (1985) is more extensive and, due to the use of in-situ optical measurements of $HNO_3$ using the same cross section as derived here, is considered to be more accurate than others obtained

at about the same time (Jourdain et al., 1982; Devolder et al., 1984). The rate coefficients of Connell and Howard were obtained in He and thus assigned an equivalent pressure in $N_2$ using the relative collision efficiency (0.38) derived by Brown et al. (1999) However, in the low pressure regime, there is only a weak dependence of $k_5$ on pressure so that constraining the





fits with data obtained in He will not introduce significant error. Equation (iii) and the parameters listed in Table 1 clearly describe our entire dataset very well. We note that re-fitting the dataset after removing the data with the largest uncertainty (i.e. that at the lowest temperatures of 208 and 217 K) has the effect of slightly *increasing* the predicted rate coefficient at the lowest temperatures. This is in the opposite direction to that which would result from the potential systematic overestimation

of $HNO_3$ concentrations at the centre of the reactor during the experiments at the lowest temperatures (see discussion in section 3.2).

**3.5 Comparison with literature and parameterisation of $k_5$ for atmospheric modelling**

In order to avoid a lengthy and repetitive discussion of previous literature results we restrict our comparison to datasets obtained in $N_2$ (or air) which are the most relevant for atmospheric chemistry. Apart from the present study we use data from

Jolly et al. (1985), Stachnik et al. (1986), Brown et al. (1999) and Carl et al. (2001). We have corrected the data of Stachnik et al. (1986) (divided each rate coefficient by 1.13) to take into account their use of absorption cross sections (Molina and Molina, 1981) of $HNO_3$ at 195 and 200 nm that were ~ 13% larger than those reported subsequently (Burkholder et al., 1993) which were obtained by more accurate methods and which agree well with the present cross section at 213.86 nm. As there are no obvious reasons to exclude any data from the studies listed above, we have performed a global fit to the

complete set of 142 separate determinations of $k_5$ in $N_2$ / air covering pressures of $\approx$ 20 -700 Torr and temperatures of 200 to 350 K. These studies were all conducted at pressures of $N_2$ of >15 Torr and do not define the rate constant well at low pressures. As described above for analysis of the present dataset, we have added rate coefficients at low pressures based on the flow tube experiments of Connell and Howard (1985) to guide the fit towards the experimental values obtained at pressures close to the low-pressure limit.

Figure 9 provides an overview of experimental room temperature rate coefficients determined in $N_2$ bath-gas by different groups. The plot also shows the parameterised rate coefficient (using the values listed in the lower part of Table 1) obtained by fitting to all datasets at all temperatures and pressures (solid line). The results are in good agreement and even the "outliers" (Jolly et al., 1985; Carl et al., 2001) agree to within 15% with the parameterisation.

In Fig. 10, we compare temperature dependent values of $k_5$ from the parameterisation with literature data. The upper panel

compares the results of the parameterisation to the present dataset, the lower panel compares it to the dataset of Brown et al. (1999) which, prior to the present data, represented the largest set of experimental rate coefficients available in the literature for $N_2$ bath gas. Figure 10 shows that the parameterisation reproduces most of the literature data obtained in $N_2$ bath gas, although the dataset of Brown et al. (1999), indicates a slightly stronger pressure dependence at most temperatures. The deviation between parameterisation and measurement is greatest for the higher temperature datasets (325, 350 K) of Brown

et al. As atmospheric $HNO_3$ is not significantly removed by reaction with OH at temperatures above $\approx$ 300 K, this does not represent a problem for use of this parameterisation for atmospheric modelling. At low temperatures (where loss of $HNO_3$ by reaction with OH is most important) the parameters capture the pressure and temperature dependence reasonably well.





Figure 11 displays the ratio of the values of $k_5$ calculated from the parameterisation presented in the present study to that derived by the IUPAC (Atkinson et al., 2004; IUPAC, 2017) and NASA-JPL (Burkholder et al., 2016) assessments, which adopted the parameters suggested by Brown et al. (1999). At pressures (< 200 mbar) and temperatures (< 240 K) typically associated with the upper troposphere and lower stratosphere, the new parameterisation results in lower values of $k_5$ with a

decrease in $k_5$ of up to 20% at 180 K. The new parameterisation reduces $k_5$ by up to 10% for the warmer (> -10 °C) temperatures associated with the lowermost troposphere.

### 3.5 Global impact of new parameterisation of $k_5$

The atmospheric impact of the new parameterisation of $k_5$ will depend on other chemical processes that generate, remove and interconvert $HNO_3$ and $NOx$ including photolysis and deposition of $HNO_3$, reaction of $NO_2$ with OH, the heterogeneous loss

of $N_2O_5$ as well as vertical and horizontal transport and was therefore examined using a global atmospheric model. We investigated the differences in model predictions of $HNO_3$ and $NOx$ mixing ratios which resulted from switching from the presently recommended parameterisation (Atkinson et al., 2004; Burkholder et al., 2016) based on the data of Brown et al. (1999) to that presented in this manuscript. The EMAC (ECHAM/MESSy Atmospheric Chemistry) model employed for this analysis is a numerical chemistry and climate simulation system (Jöckel et al., 2006; Jöckel et al., 2010) using  the 5th

generation European Centre Hamburg general circulation model (ECHAM5, Roeckner et al. (2006)) as core atmospheric general circulation model. For the present study we applied EMAC (ECHAM5 version 5.3.02, MESSy version 2.53.0) in the T42L47MA-resolution, i.e. with a spherical truncation of T42 (corresponding to a quadratic Gaussian grid of approx. 2.8 by 2.8 degrees in latitude and longitude) with 47 vertical hybrid pressure levels up to 0.01 hPa. The model has been weakly nudged in spectral space, nudging temperature, vorticity, divergence and surface pressure (Jeuken et al., 1996). The chemical

mechanism scheme adopted (MOM, Mainz Organic Mechanism) includes oxidation of isoprene, saturated and unsaturated hydrocarbons, including terpenes and aromatics (Cabrera-Perez et al., 2016; Lelieveld et al., 2016). Further, tracer emissions and model set-up are similar to the one presented in Lelieveld et al. (2016). EMAC model predictions have been evaluated against observations on several occasions (Pozzer et al., 2010; de Meij et al., 2012; Yoon and Pozzer, 2014): For additional references, see http://www.messy-interface.org. For this study, EMAC was used in a chemical-transport model (CTM mode)

(Deckert et al., 2011), i.e., by disabling feedbacks between photochemistry and dynamics. Two years were simulated (2009-2010), with the first year used as spin-up time. The lower panel of Fig. 11 illustrates the latitude and altitude dependent changes (zonal and annual averages) in the modelled rate coefficient $k_5$. Increases in $k_5$ (up to 8%) are seen throughout the lower stratosphere, whereas the warmer temperatures and higher pressures of the troposphere generally result in a decrease in $k_5$.

The results of the model simulations are expressed in Fig. 12 as percentage changes in the zonally and annual averaged $HNO_3$ (upper panel) and $NOx$  mixing ratios (lower panel) as a function of latitude and altitude. The largest relative changes in $HNO_3$ are found in the tropical upper troposphere (+ 4%) with a corresponding reduction in $NOx$ of 2%. The largest relative change in $NO_X$ (+4%) is predicted to be in the lower stratosphere (18-23 km) at mid- and high-latitudes. As $HNO_3$



and NO*x* are impacted in different directions by changing $k_5$, the HNO$_3$ / NO*x* ratio will be most strongly affected. The predicted, global relative change in the HNO$_3$ / NO*x* ratio (Fig. S4 of the supplementary information) was found to be -6% in the tropical upper troposphere and +6% in the lower stratosphere. Given the importance of HNO$_3$ to NO*x* partitioning in governing rates of photochemical ozone production, the impact of the changes in $k_5$ are significant.

## 4 Conclusions

We have obtained a large number of rate coefficients ($k_5$) for the reaction between OH and HNO$_3$, the associated uncertainty of which was minimised through in-situ optical measurements of HNO$_3$ by both absorption and fluorescence methods and detailed analysis of impurities. The pressure and temperature dependent rate coefficients confirm the results of an earlier

study (Brown et al., 1999), which found a strong increase in the pressure dependence of $k_5$ at low temperatures. The rate coefficients were combined with previous results to give a global parameterisation that is applicable to the Earth's atmosphere and which results in latitude and altitude dependent changes in the HNO$_3$ / NOx ratio of between +6 and -6%.

### Acknowledgements

We thank the Deutsche Forschungsgemeinschaft (DFG) for partial financial support (CR 246/2-1). We are grateful to The Chemours Company (CH) for providing a sample of the FEP suspension used to coat the reactor.

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





**Table 1.** Parameters for calculation of $k_5$ using equation (iii)

| | Present dataset (in $N_2$) plus parameterisation[a] of low pressure data |
|---|---|
| $k_0$ | $1.58 \times 10^{-14}$ exp $(451/T)$ cm$^3$ molecule$^{-1}$ s$^{-1}$ |
| $k_\Delta$ | $1.23 \times 10^{-16}$ exp $(1854/T)$ cm$^3$ molecule$^{-1}$ s$^{-1}$ |
| $k_c$ | $8.46 \times 10^{-32}$ exp $(525/T)$ cm$^6$ molecule$^{-2}$ s$^{-1}$ |
| | All data in $N_2$ bath gas[b] plus parameterisation[a] of low P data |
| $k_0$ | $1.32 \times 10^{-14}$ exp $(527/T)$ cm$^3$ molecule$^{-1}$ s$^{-1}$ |
| $k_\Delta$ | $9.73 \times 10^{-17}$ exp $(1910/T)$ cm$^3$ molecule$^{-1}$ s$^{-1}$ |
| $k_c$ | $7.39 \times 10^{-32}$ exp $(453/T)$ cm$^6$ molecule$^{-2}$ s$^{-1}$ |

[a]$k_5 = 2 \times 10^{-14}$ exp $(430/T)$ cm$^3$ molecule$^{-1}$ s$^{-1}$ at M $= 2.5 \times 10^{16}$ molecule cm$^{-3}$.

[b]140 data points in $N_2$ including this work, Stachnik et al. (1986) and Brown et al. (1999).



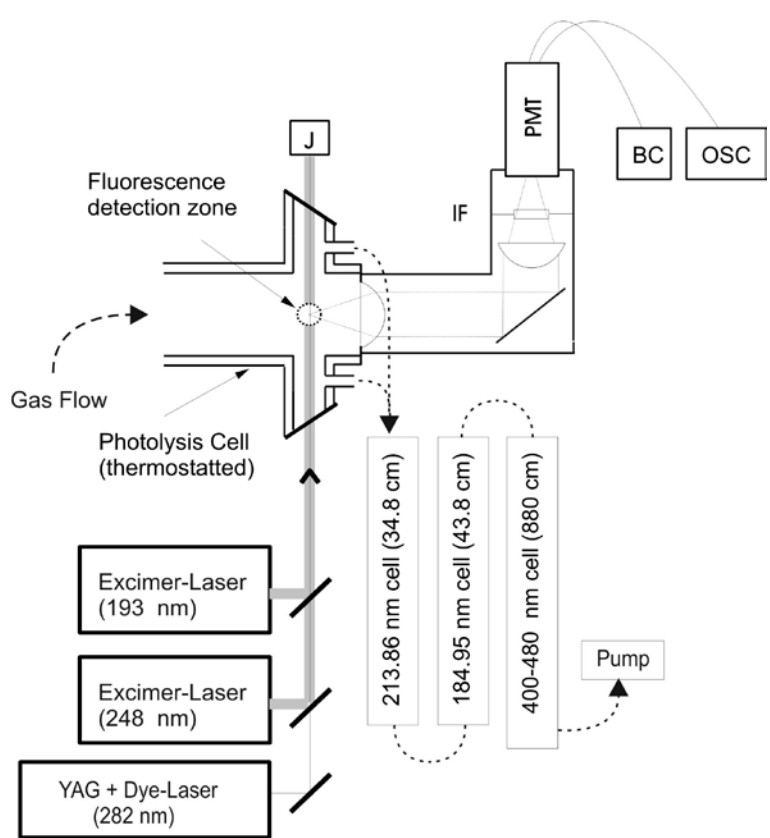

**Figure 1:** PLP-LIF setup. MFC = mass flow controller, PMT = photomultiplier tube, J = Joule meter, BC = box-car data acquisition, OSC = oscilloscope, IF = interference filter (309 ± 5 nm). Capacitance manometers monitored the pressure inside the reactor and the three optical absorption cells.





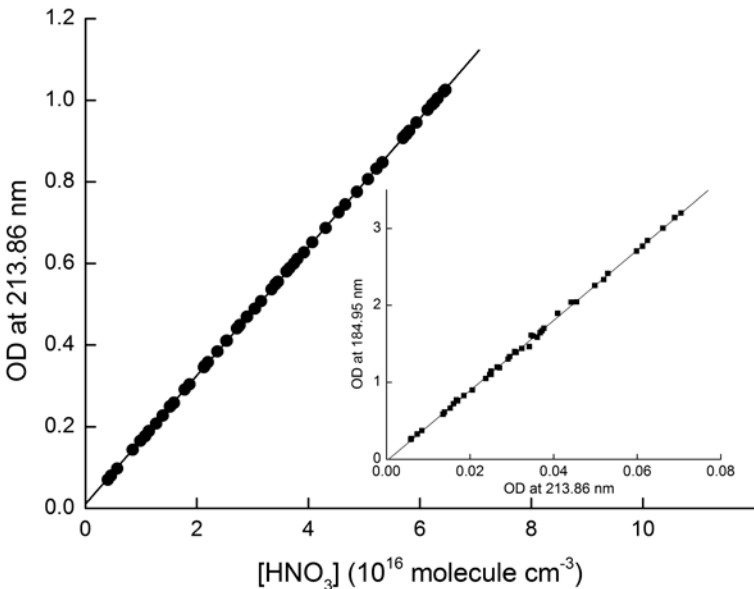

**Figure 2**: Beer-Lambert plot used for the calculation of the $HNO_3$ absorption cross section at 213.86 nm, $\sigma_{213.86\ nm}$. The inset shows the linear relation between optical densities (OD) at 184.95 nm and 213.86 nm.





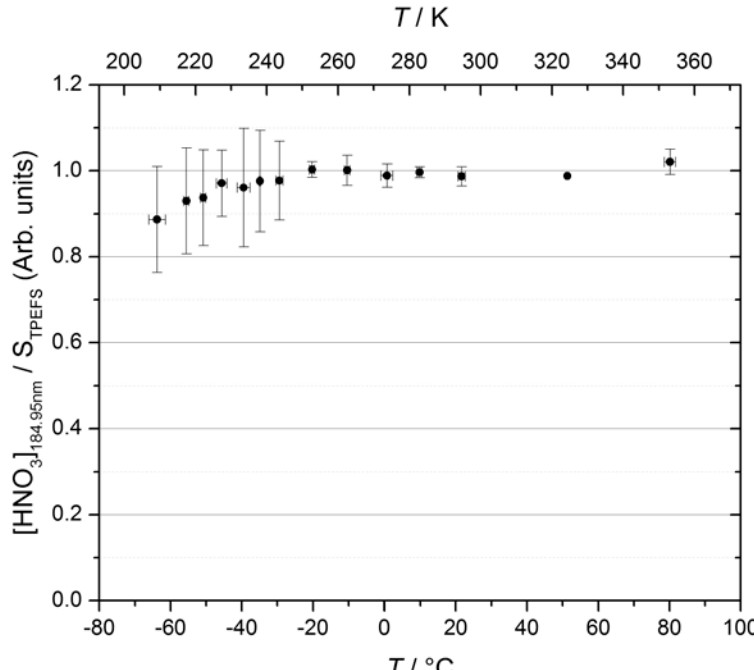

**Figure 3**. TPEFS detection of $HNO_3$ in the center of the reactor at different temperatures. The TPEFS signal ($S_{TPEFS}$) has been normalised to the $HNO_3$ concentration. The error bars indicate uncertainty derived from correction to the TPEFS signal due to quenching of fluorescence and absorption by $HNO_3$.



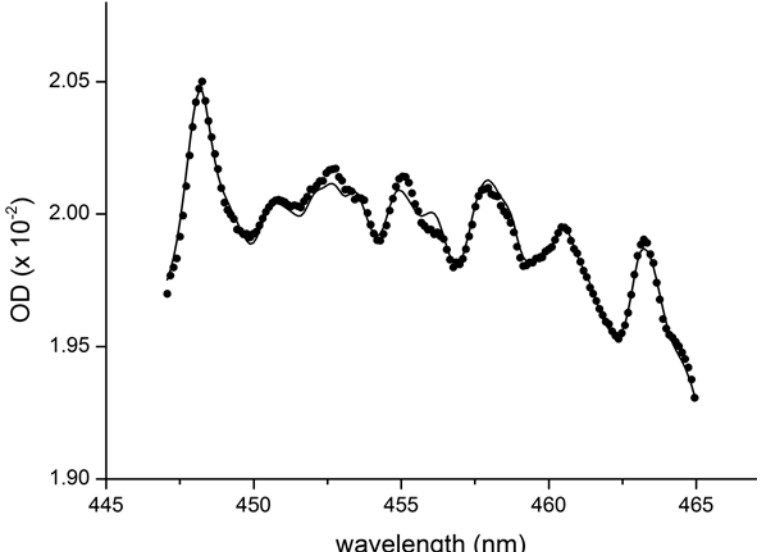

**Figure 4**. NO$_2$ impurity measurement (experiment at 298 K, 18 Torr) using the ternary mixture. The dotted line is the experimental NO$_2$ optical density (OD); the solid line is the fitted optical density using a reference spectrum (Bogumil et al., 2003). In this measurement, the NO$_2$ impurity corresponded to 0.01% of the HNO$_3$ concentration.


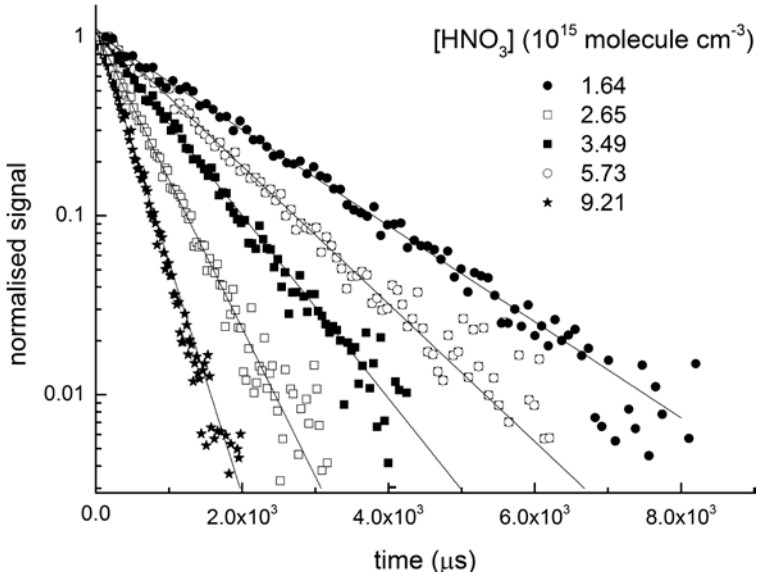

**Figure 5**. Temporal decay of the OH-LIF signal at 242 K and 50 Torr N$_2$.





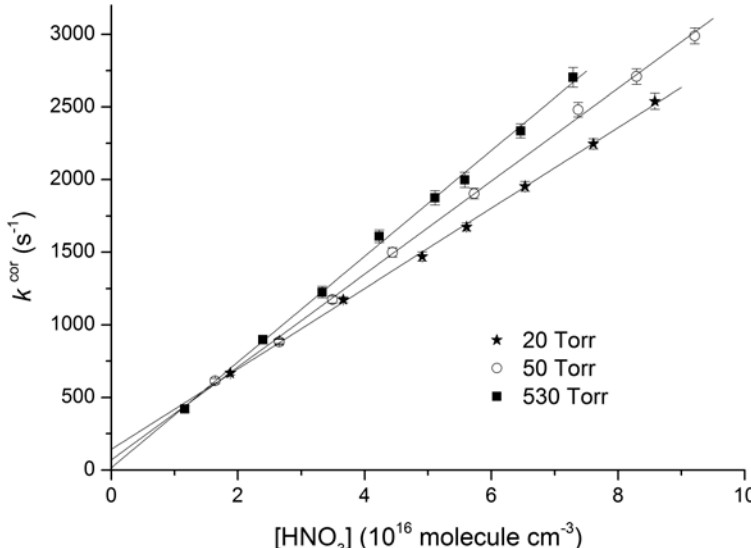

**Figure 6**. Corrected pseudo-first-order rate coefficients, $k^{cor}$, plotted as a function of the HNO$_3$ concentration. The data were obtained at 242 K and at three different pressures.





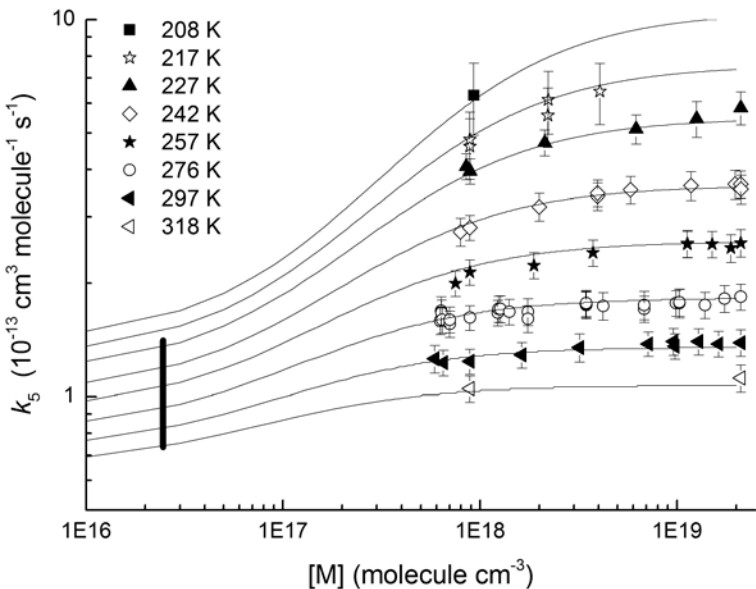

**Figure 7**. Rate coefficients $k_5$ as a function of pressure and temperature from this work (error bars represent total uncertainty). The solid lines are fits to the entire dataset with expression (iii). Data at 246 and 239 K are omitted to preserve clarity of presentation. The vertical line at $[M] = 2.5 \times 10^{16}$ molecule $cm^{-3}$ (220-330 K) represents the dataset used to guide the fit at low pressures and is given by the expression $k_5 = 2 \times 10^{-14}$ exp $(430/T)$ $cm^3$ molecule$^{-1}$ s$^{-1}$.





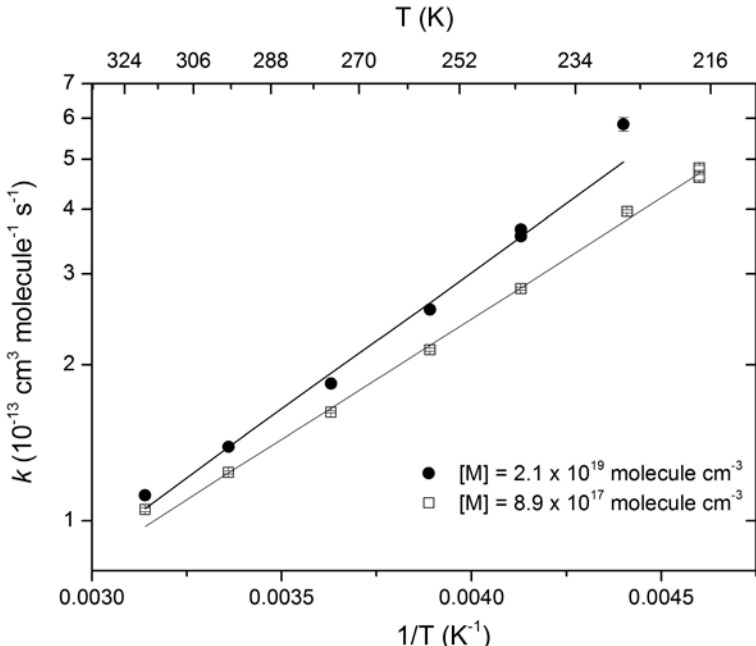

**Figure 8**. Temperature dependent rate coefficients at two constant bath gas concentrations [M]. The statistical uncertainties of the experimental data are within the symbol size. The solid lines are Arrhenius fits to the data (see text).





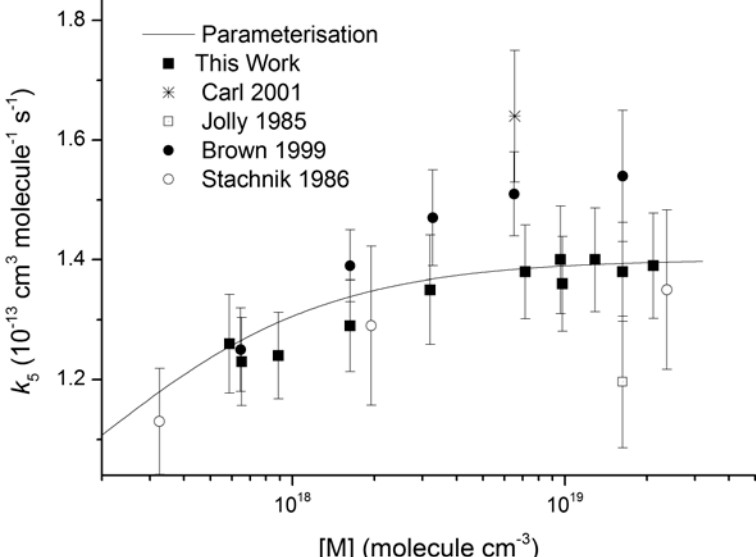

**Figure 9**. Comparison of room temperature rate coefficients ($k_5$) obtained in nitrogen bath gas. The error bars on our data points represent total uncertainty. The solid line are values of $k_5$ at 297 K from the parametrization obtained by fitting to the temperature and pressure dependent datasets from Brown et al. (1999), Stachnik et al. (1986), and this work.



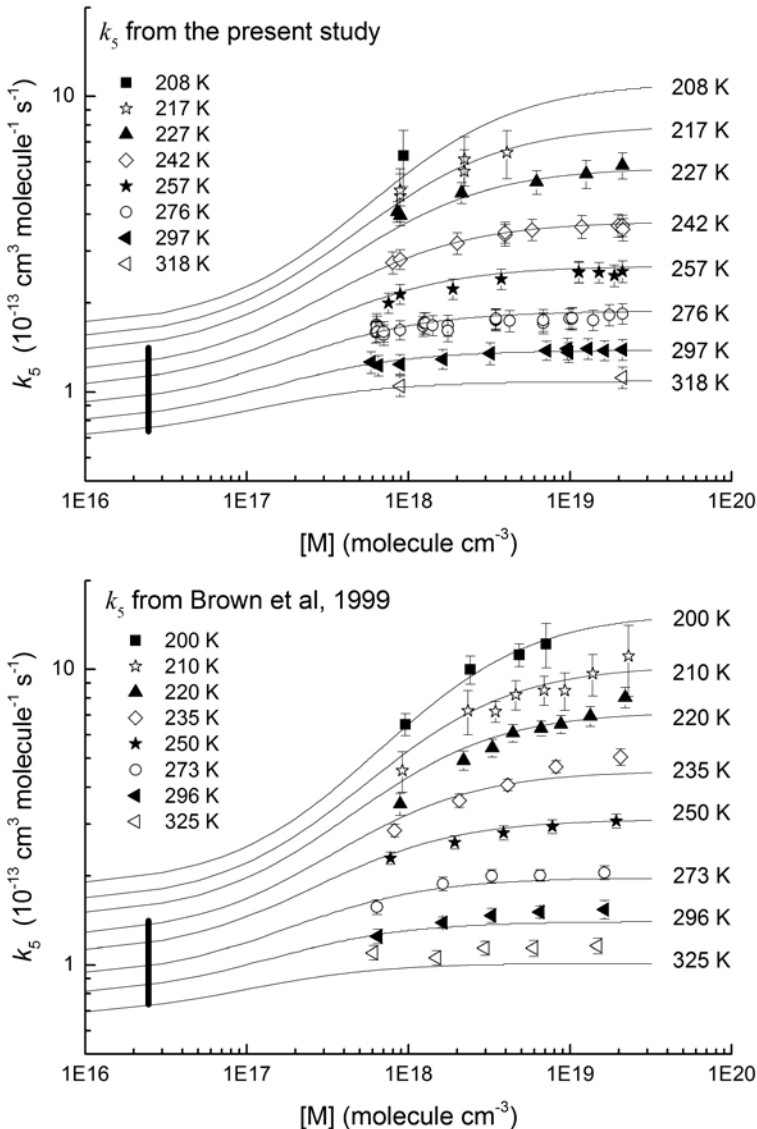

**Figure 10**. Rate coefficients $k_5$ as a function of pressure and temperature from Brown et al. (1999) (lower panel) and the present study (upper panel, error bars represent total uncertainty). The solid lines were derived from the parameters listed in Table 1 (lower panel). The vertical line at M = $2.5\times10^{16}$ molecule cm$^{-3}$ (220-330 K) represents the dataset used to constrain the fit at low pressures and is given by the expression $k_5 = 2\times10^{-14}$ exp $(430/T)$ cm$^3$ molecule$^{-1}$ s$^{-1}$.



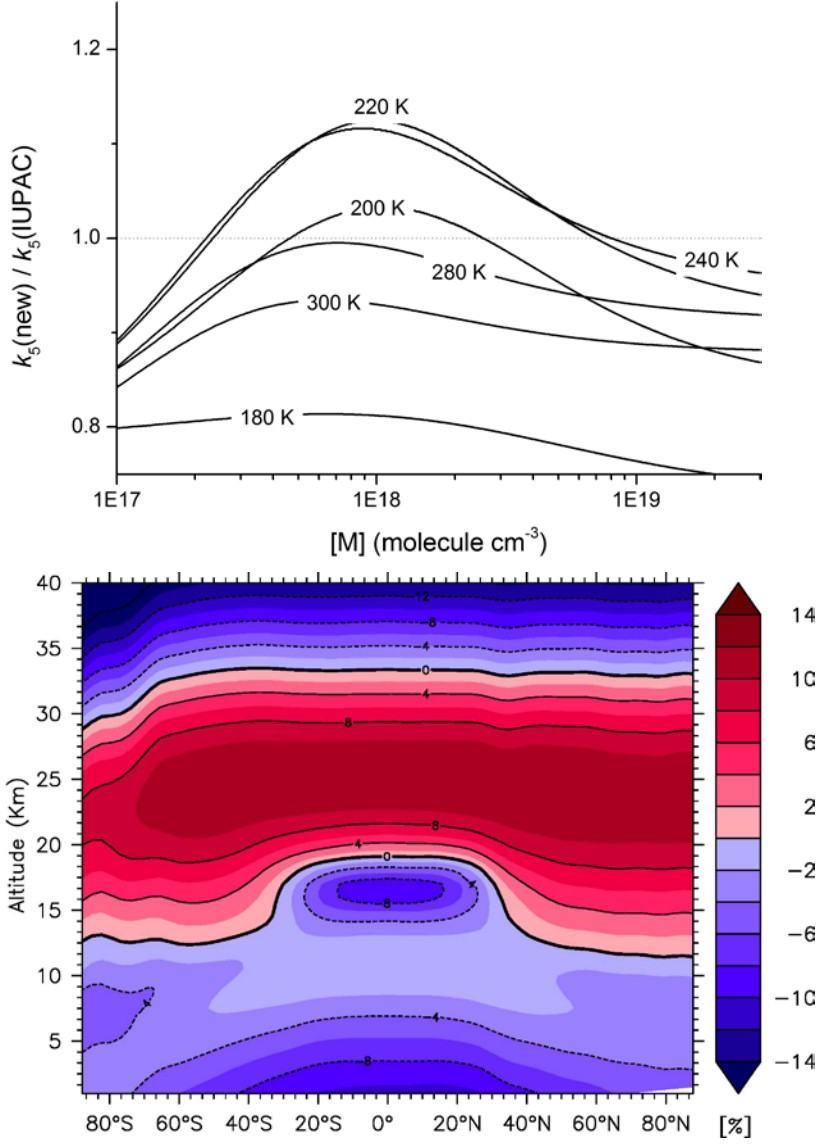

**Figure 11**. Upper panel: Ratio of rate coefficients obtained by combining results from this work with selected datasets from the literature, $k_5$(new), to those presently recommended by IUPAC, $k_5$(IUPAC). Lower panel: percentage change in rate coefficient $k_5$ calculated using annually and zonally averaged temperatures and pressure in the EMAC model.



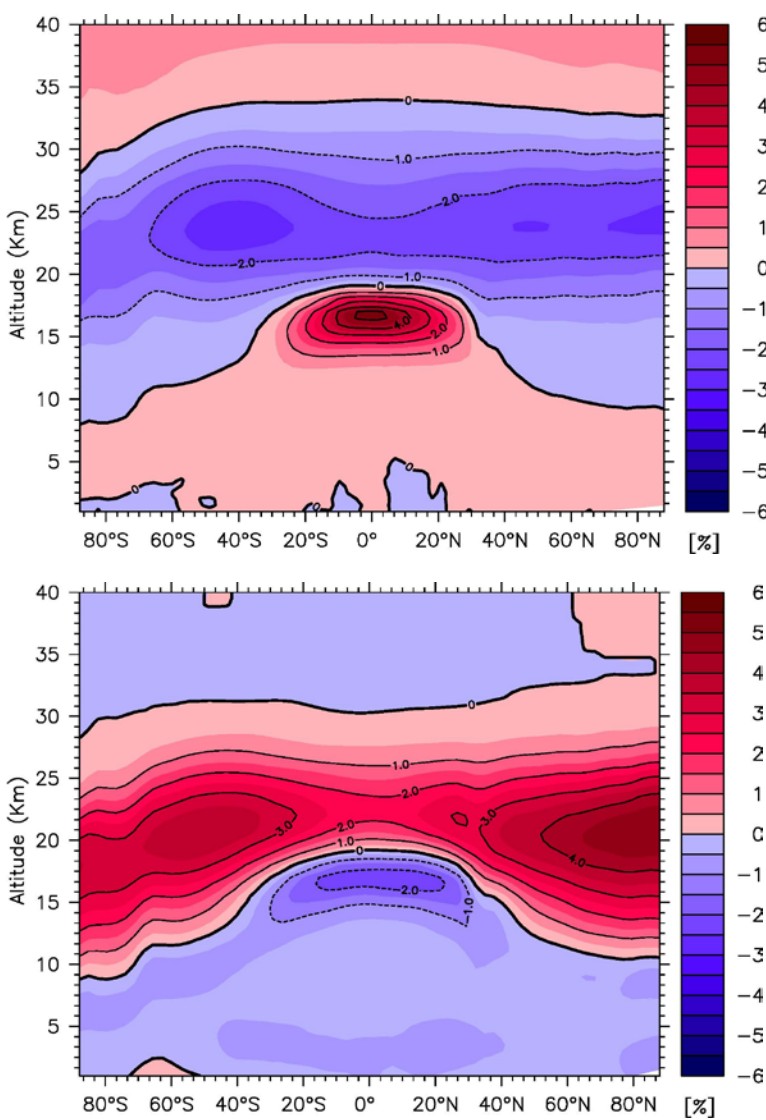

**Figure 12**. Impact of the results from this work in a global atmospheric chemistry model (EMAC). The contours indicate the percentage change in the predicted mixing ratios (zonal and annual average) of $HNO_3$ (upper panel) and NOx (lower panel), which result from the new parameterization of $k_5$.