# Peer review of "Temperature (208-318 K) and pressure (18-696 Torr) dependent rate coefficients for the reaction between OH and HNO3."

_Atmospheric Chemistry and Physics, 2017_

## Referee Comment (RC1) · P. Seakins (Referee) · 30 Oct 2017

Dulitz et al. Review

This paper reports a very careful study on the reaction of the OH radical with nitric acid ($HNO_3$). Technically, this is a very difficult reaction to study, but it is important to generate precise and accurate rate coefficients as this reaction is vital in controlling $HNO_3$/NOx ratios. Due to the atmospheric relevance, this material is appropriate for publication in ACP.

I have no significant concerns about the paper, which I strongly recommend for publication, however, as well as some minor details listed below, there are a couple of issues that I would like the authors to consider for the final publication.

1. The reaction shows interesting temperature and pressure behaviour. Although ACP is not the vehicle for a detailed discussion of the fundamental mechanisms, I would ask the authors to consider including a **brief** rationale of the physical model for two reasons. Firstly, it would illustrate to potential users (modellers primarily?) the issues that need to be considered in laboratory studies, not just the technical issues as detailed here, but also the theoretical understanding. The importance of conveying such information to a wider audience was highlighted in a recent article by Burkholder et al. in ES&T. Secondly, and this is related to the next point, understanding the mechanism allows for a better assessment of whether the parameterizations are valid outside of the measurement regime.

2. As mentioned above, the data have been parameterized, which will be useful to modellers, but there is no mention of the uncertainty in the parameterization constants or on the validity of the parameterization outside the experimental range. The biggest differences in the ratio of rate coefficients reported in Fig 11 a are for 180 K, significantly below the current measurements and those of Brown et al. Uncertainties in parameterizations can be difficult to present as the parameters can be highly correlated and simple error ranges may underestimate the total uncertainty. However, some discussion needs to be presented on both of these issues (uncertainty and validity beyond experimental range).

Minor Points

1. Give values of $k_5$ in the abstract – at least room temperature and atms pressure.
2. page 2 – Give an example of the magnitude of the measured:modelled HNO3 concentrations and $HNO_3$:NOx ratios.
3. page 3 – Give typical laser fluence (or range of fluence) in mJ cm$^{-2}$ pulse$^{-1}$.
4. page 8, Section 3.3. I would suggest re-titling as Impurities and Secondary Reactions.
5. page 9 – Clarify the results of the air vs $N_2$ experiment. A difference within 4% is reported. My assumption would be that therefore no significant difference between $k_5$ reported in air vs $N_2$, but this should be clarified.

6. page 10, Terminology – $k_0$ is not the best term to describe the pressure independent term in $k_5$. $k_0$ has a specific meaning within Lindemann-Hinshelwood theory and therefore there is potential for confusion.
7. page 12, Fig 11 – The text at the top of p12 refers to differences between current parameterization and IUPAC and JPL, but only IUPAC data presented in the Figure.
8. p13 – The final sentence of section 3.5 needs expanding to make it clear why a 6% change is significant
9. References – Jolly et al. CPL, should be Chem. Phys. Lett. Several references need correcting for subscripts and capitals (e.g. Uv)
10. Figures – Would suggest more use of colour for Figs 4 onwards. Because the authors have chosen symbols etc carefully, most figs work fine in b/w, but most could be enhanced with a bit of colour.
11. Figs 11 and 12 – Captions need a bit more detail. e.g. Fig 11 'change in rate coefficient k5' – compared to what? Is it the new parameterization including this work and literature or just this work?

---

## Referee Comment (RC2) · Anonymous Referee #2 · 22 Nov 2017

The authors report rate coefficients for the reaction of OH with HNO3 over a wide range of temperature and pressure. The reaction is of atmospheric significance as it directly influences NOx levels and the NOx/HNO3 ratio. This is an excellent paper, detailing a careful and thorough study of the reaction, and I have no hesitation in recommending its publication (subject to consideration of a few minor points made below). In particular, I applaud the efforts made by the authors to measure as best as possible in situ the HNO3 concentrations, as well as the levels of numerous possible interferences, to provide what is a very reliable set of data.

A few minor comments:

[Figure]

Page 4, line 20 – 'concentration at the centre . . .'

You might mention in some way in the caption to Figure 2 that the pathlengths for the two cells are different, so that the ratio of the OD's are not equal to the ratio of the cross sections obtained.

Again, I applaud the efforts made to quantify HNO3 levels via TPEFS. But, is it not the case that the TPEFS is calibrated by measuring the [HNO3] downstream, making the argument partially circular? I understand that the agreement over a large range of temperatures (with a small possible downturn at low T) is very re-assuring, but could it be that there is a little bit of loss occurring at all temperatures that puts some kind of systematic bias to the whole dataset? If I am correct in this assessment, maybe just one sentence to clarify assumptions made here would be warranted.

Page 7, line 3 - missing a period after (1985).

Page 9, line 16 – 'determinations' should be plural.

Page 12 and Figure 11 – Can the authors say anything about why the difference between 'old' and 'new' suddenly increases at 180 K?

---

## Author Comment (AC1) · 10 Jan 2018

The following contains the comments of the three referees (black), our replies (blue) indicating changes that will be made to the revised document (red).

| Referee: Paul Seakins |
|---|
| Comment
This paper reports a very careful study on the reaction of the OH radical with nitric acid (HNO3). Technically, this is a very difficult reaction to study, but it is important to generate precise and accurate rate coefficients as this reaction is vital in controlling HNO3/NOx ratios. Due to the atmospheric relevance, this material is appropriate for publication in ACP. I have no significant concerns about the paper, which I strongly recommend for publication, however, as well as some minor details listed below, there are a couple of issues that I would like the authors to consider for the final publication.

Reply
We thank Dr. Seakins for this careful review and very positive assessment of our manuscript |
| Comment
1. The reaction shows interesting temperature and pressure behaviour. Although ACP is not the vehicle for a detailed discussion of the fundamental mechanisms, I would ask the authors to consider including a brief rationale of the physical model for two reasons. Firstly, it would illustrate to potential users (modellers primarily?) the issues that need to be considered in laboratory studies, not just the technical issues as detailed here, but also the theoretical understanding. The importance of conveying such information to a wider audience was highlighted in a recent article by Burkholder et al. in ES&T. Secondly, and this is related to the next point, understanding the mechanism allows for a better assessment of whether the parameterizations are valid outside of the measurement regime.

Reply
The model used to parameterise the data was developed by Lamb et al. and is consistent with a reaction that proceeds via formation of an association complex, the fate of which is already described at the bottom of page 2. As this has been discussed many times in various publications dealing with the kinetics of this reaction (which are all cited) we see no real benefit in repeating this here. Note that we have modified the form of the equation (mathematically equivalent) to be consistent with its previous usage.
We have added text to encourage high level theoretical studies of this reaction that provide a working theoretical framework for confident prediction of coefficients outside of the experimentally accessible temperature range:
High-level theoretical studies of the title reaction and experimental studies at temperatures not accessible by standard methods (e.g. using Laval nozzle expansions) would be useful to provide a working theoretical framework and a more physical parameterisation of the data, which in turn allow for confident prediction of rate coefficients outside of the presently available temperature range. |
| Comment
2. As mentioned above, the data have been parameterized, which will be useful to modellers, but there is no mention of the uncertainty in the parameterization constants or on the validity of the parameterization outside the experimental range. The biggest differences in the ratio of rate coefficients reported in Fig 11 a are for 180 K, significantly below the current measurements and those of Brown et al. Uncertainties in parameterizations can be difficult to present as the parameters can be highly correlated and simple error ranges may underestimate the total uncertainty. However, some discussion needs to be presented on both of these issues (uncertainty and validity beyond experimental range).

Reply
Although 180 K is "only" 23 K cooler than the lowest temperature of Brown et al., this is a valid point. We have added text and a Figure that addresses the uncertainty of the parameterisation within and |

outside the range of temperatures covered experimentally:

At low temperatures (for which the loss of $HNO_3$ by reaction with OH is most important) the parameters capture the pressure and temperature dependence reasonably well. The agreement is illustrated in Fig. S4 of the supplementary information which plots the measured and parameterised rate coefficient against each other. This plot has a slope of $0.99 \pm 0.01$ with $R^2 = 0.99$ and indicates only slight deviation from a linear relationship at the highest temperatures. The confidence limits of the fit-line are within the scatter of the experiments, suggesting that, between $\approx 200$ and 290 K) the parameterisation does not introduce uncertainty beyond that associated with the experiments. We thus estimate an overall uncertainty of $\approx 15$ % in the parameterised rate coefficients within the range of temperatures studied experimentally and those relevant for the OH-initiated removal of atmospheric $HNO_3$, i.e. $200 - 290$ K. Extrapolation to temperatures lower than those covered experimentally may result in larger uncertainty as becomes apparent when comparing the new and old parameterisations.

Comment

1. Give values of k5 in the abstract – at least room temperature and atms pressure.

Reply

Temperatures close to room temperature and 1 bar are found in the planetary boundary layer, where HNO3 is lost by deposition and not (significantly) by reaction with OH. The focus of this paper is on the atmospherically important low temperature kinetics of the reaction and listing k5 at room temperature and I bar in the abstract (conditions where none of the present experiments were conducted) would not provide useful information.

Comment

2. page 2 – Give an example of the magnitude of the measured:modelled HNO3 concentrations and HNO3:NOx ratios.

Reply

The abundance of HNO3 and the HNO3:NOx ratio is extremely variable. There are large horizontal and vertical gradients and great differences in e.g. the boundary layer, the lower troposphere and the stratosphere. There is no "typical" value and a single example (or even two or three) would not inform the interested reader. A discussion of this is covered in many papers that deal with the measurements of these trace gases in the atmosphere, several of which we already cite.

Comment

3. page 3 – Give typical laser fluence (or range of fluence) in mJ cm-2 pulse-1 .

Reply

This information is now given in section 2.2:

The exciplex laser fluence (2-4 mJ cm$^{-2}$ pulse$^{-1}$) was adjusted so that the $[HNO_3]/[OH]$ ratio was always $> 10^4$ ensuring first-order conditions and suppressing interfering secondary OH chemistry (see section 3.3).

Comment

4. page 8, Section 3.3. I would suggest re-titling as Impurities and Secondary Reactions.

Reply

Section 3.3 has been re-titled:

3.3 Impurities and secondary chemistry

Comment

5. page 9 – Clarify the results of the air vs N2 experiment. A difference within 4% is reported. My assumption would be that therefore no significant difference between k5 reported in air vs N2, but this should be clarified.

Reply

Correct. We have added:

Within experimental uncertainty and the temperature and pressure range studied, there is thus no significant dependence of $k_5$ on use of $N_2$ or air bath gases.

Comment

6. page 10, Terminology – k0 is not the best term to describe the pressure independent term in k5. k0 has a specific meaning within Lindemann-Hinshelwood theory and therefore there is potential for confusion.

Reply

We have replaced $k_0$ with k.

Comment

7. page 12, Fig 11 – The text at the top of p12 refers to differences between current parameterization and IUPAC and JPL, but only IUPAC data presented in the Figure.

Reply

AS both IUPAC and JPL adopt the same parameters, the recommendations are identical. We have modified Fig. 11 and the caption to mention both IUPAC and JPL.

Comment

8. p13 – The final sentence of section 3.5 needs expanding to make it clear why a 6% change is significant

Reply

We agree that this is a rather ambiguous statement. However, it is difficult to be more quantitative as no firm definition of what represents a significant change in e.g. the amounts of photochemically generated ozone exists. Instead we now indicate that, in terms of sensitivity to rate constant changes, the reaction between OH and HNO3 is in the top ten reactions that impact on the global O3 burden and cite a very recent paper (Newsome and Evans, 2017) in ACPD:  Given the importance of $HNO_3$ to $NOx$ partitioning in governing rates of photochemical ozone production (Newsome and Evans, 2017) and the important role of the OH + $HNO_3$ reaction, the impact of the changes in $k_5$ are significant.

Comment

9. References – Jolly et al. CPL, should be Chem. Phys. Lett. Several references need correcting for subscripts and capitals (e.g. Uv)

Reply

Corrections made

Comment

10.Figures – Would suggest more use of colour for Figs 4 onwards. Because the authors have chosen symbols etc carefully, most figs work fine in b/w, but most could be enhanced with a bit of colour.

Reply

Figures 7 and 10 have now been reproduced with colour.

Comment

11.Figs 11 and 12 – Captions need a bit more detail. e.g. Fig 11 'change in rate coefficient k5' – compared to what? Is it the new parameterization including this work and literature or just this work?

Reply

We have clarified this by writing (in the caption):

Figure 11

Lower panel: Change in rate coefficient $k_5$(new) compared to previous IUPAC and JPL evaluations. The percentage change was calculated using annually and zonally averaged temperatures and

pressure in the EMAC model.

Figure 12
The contours indicate the percentage change in the predicted mixing ratios (zonal and annual average) of HNO3 (upper panel) and NOx (lower panel), which result from the new parameterization of k5 compared to the IUPAC and JPL recommended values.

**Referee 2**

Comment
The authors report rate coefficients for the reaction of OH with HNO3 over a wide range of temperature and pressure. The reaction is of atmospheric significance as it directly influences NOx levels and the NOx/HNO3 ratio. This is an excellent paper, detailing a careful and thorough study of the reaction, and I have no hesitation in recommending its publication (subject to consideration of a few minor points made below). In particular, I applaud the efforts made by the authors to measure as best as possible in situ the HNO3 concentrations, as well as the levels of numerous possible interferences, to provide what is a very reliable set of data.

Reply
We thank referee 2 for this careful review and very positive assessment of our manuscript

Comment
Page 4, line 20 – 'concentration at the centre . . .'

Reply
Correction made

Comment
You might mention in some way in the caption to Figure 2 that the pathlengths for the two cells are different, so that the ratio of the OD's are not equal to the ratio of the cross sections obtained.

Reply
In the caption to Figure 2 we now write:
The inset shows the linear relation between optical densities (OD) at 184.95 nm (optical path-length = 43.8 cm) and 213.86 nm (optical path-length = 34.8 cm).

Comment
Again, I applaud the efforts made to quantify HNO3 levels via TPEFS. But, is it not the case that the TPEFS is calibrated by measuring the [HNO3] downstream, making the argument partially circular? I understand that the agreement over a large range of temperatures (with a small possible downturn at low T) is very re-assuring, but could it be that there is a little bit of loss occurring at all temperatures that puts some kind of systematic bias to the whole dataset? If I am correct in this assessment, maybe just one sentence to clarify assumptions made here would be warranted.

Reply
The point of the TPEFS is to look for gradients in HNO3 concentrations across the diameter of the reactor and not to check the concentration in an absolute sense. Our kinetic data do not rely on calibrations of the TPEFS signal in any way. The only assumption made is that there is no gradient in HNO3 concentrations at warmer temperatures where its partitioning to the reactor walls is less important.

Comment
Page 7, line 3 - missing a period after (1985).
Page 9, line 16 – 'determinations' should be plural.

Reply
Corrections made

Comment
Page 12 and Figure 11 – Can the authors say anything about why the difference between 'old' and 'new' suddenly increases at 180 K?

Reply
At the lowest temperatures (and pressures distinct from the low pressure limiting regime) the reaction will proceed mainly via the stabilised association complex and the rate coefficient is influenced most by the values of $k_c$ at different temperatures. Indeed, the largest differences between the present recommendation and the IUPAC and JPL parameters (based on Brown et al., 1999) is found in the terms for $k_c$. $k_c$ (new) = 8.46 × 10$^{-32}$ exp (525/$T$) cm$^6$ molecule$^{-2}$ s$^{-1}$, and $k_c$ (old) = 6.51 × 10$^{-34}$ exp (1335/$T$) cm$^6$ molecule$^{-2}$ s$^{-1}$. We have added text to illustrate this:
At pressures (< 200 mbar) and temperatures (< 240 K) typically associated with the upper troposphere and lower stratosphere, the new parameterisation results in lower values of $k_5$ with a decrease in $k_5$ of up to 20% at 180 K, which is beyond the range of temperatures studied experimentally.  As the parameters for $k_\Delta$ and $k$ are similar to those previously recommended, the large difference at 180 K most likely reflects changes in the temperature dependence of $k_c$ with the older value represented by 6.51 × 10$^{-34}$ exp (1335/$T$) cm$^6$ molecule$^{-2}$ s$^{-1}$ compared to  the present value of 8.46 × 10$^{-32}$ exp (525/$T$) cm$^6$ molecule$^{-2}$ s$^{-1}$.